# Effect of ovarian stimulation on the expression of piRNA pathway proteins

**Ismail Sari**[ID][1]*, **Erkan Gumus**[2], **Ahmet Sevki Taskiran**[3], **Lale Karakoc Sokmensuer**[4]

**1** Department of Medical Biochemistry, Faculty of Medicine, Nigde Omer Halis Demir University, Nigde, Turkey, **2** Department of Histology and Embryology, Faculty of Medicine, Aydın Adnan Menderes University, Aydın, Turkey, **3** Department of Physiology, Faculty of Medicine, Sivas Cumhuriyet University, Sivas, Turkey, **4** Department of Histology and Embryology, Faculty of Medicine, Hacettepe University, Ankara, Turkey

* smlsr@hotmail.com

**Data Availability Statement:** All relevant data are within the manuscript and its Supporting Information files.

## Abstract

PIWI-interacting RNAs (piRNAs) play an important role in gametogenesis, fertility and embryonic development. The current study investigated the effect of different doses of pregnant mare serum gonadotrophin/human chorionic gonadotrophin (PMSG/hCG) and repeated ovarian stimulation (OS) on the expression of the *Mili*, *Miwi*, *Mael*, *Tdrd1*, *Tdrd9*, qnd *Mitopld* genes, which have crucial roles in the biogenesis and function of piRNAs. Here, we found that after treatment with 7.5 I.U. PMSG/hCG and two repeated rounds of OS, both the mRNA and protein levels of *Tdrd9*, *Tdrd1* and *Mael* showed the greatest decrease in the ovarian tissue, but the plasma $E_2$ levels showed the strongest increases (p<0.05). However, we found that the *Mitopld*, *Miwi* and *Mili* gene levels were decreased significantly after treatment with 12.5 I.U. PMSG/hCG. Our results suggested that exogenous gonadotropin administration leads to a significant decrease in the expression of the *Mili*, *Miwi*, *Mael*, *Tdrd1*, *Tdrd9* and *Mitopld* genes, which are critically important in the piRNA pathway, and the changes in the expression levels of *Tdrd9*, *Tdrd1* and *Mael* may be associated with plasma $E_2$ levels. New comprehensive studies are needed to reduce the potential effects of OS on the piRNA pathway, which silences transposable elements and maintains genome integrity, and to contribute to the safety of OS.

## Introduction

Ovarian stimulation (OS) with exogenous gonadotropin injections has been used for many years as a method for increasing oocytes in animal and humans. Gonadotropins are also used in infertility treatments. Although considerable progress in *in vitro* fertilization (IVF) has been achieved in recent years, the pregnancy rate per embryo transferred is still low [1]. Many studies comparing natural and stimulated ovarian cycles have indicated some detrimental effects of gonadotropin stimulation, and there may be a relationship between treatment with gonadotropins and a low pregnancy rate. Furthermore, increased chromosomal abnormalities were found in gonadotropin-treated mice and rats, suggesting that genetic factors may be implicated in embryonic mortality [2–4]. Since such potential abnormalities in embryos and

**Funding:** This work was supported by TUBITAK (The Scientific and Technological Research Council of Turkey) under grand number 116s674.

**Competing interests:** The authors declare that there is no conflict of interest.

offspring are elicited by OS, it is necessary to determine the underlying defects associated with this procedure. Because OS is essential in the treatment of infertility, elucidation of the exact mechanisms responsible for these detrimental effects of OS is urgently needed to increase the success of IVF.

In recent years, many studies have shown that small noncoding RNAs (sncRNAs), including microRNAs (miRNAs), small endogenous interfering RNAs (siRNAs), and piwi-interacting RNAs (piRNAs), have crucial roles in reproductive functions [5]. piRNAs have a special function in reproductive biology among these sncRNAs. piRNAs are a novel class of noncoding small single-stranded RNAs abundant in the germline across animal species. Previous studies have demonstrated that piRNAs play crucial roles in gametogenesis, tumorigenesis, epigenetic regulation, germline development, transposon silencing and regulation of gene expression [6]. Host-defense mechanisms against transposable elements are essential to protect genome information. The piRNA pathway, which contains PIWI proteins, piRNAs and proteins that have a role in piRNA biogenesis, can maintain genome integrity by transposon silencing. Thus, piRNAs are also considered "the guardians of the genome". Furthermore, it has also been shown that piRNAs and PIWI proteins are expressed and function in somatic cells [7, 8].

The main effector complex of the piRNA pathway is named the piRNA-inducing silencing complex (RISC), which includes piRNAs and PIWI proteins. In this complex, piRNAs guide PIWI-clade proteins to complementary target RNAs to initiate silencing. The PIWI proteins belong to the argonaute superfamily and have highly conserved PIWI, PAZ, and MID domains. The PAZ domain includes an oligonucleotide-binding fold that interacts with the 3' ends of piRNAs, and the PIWI domain is homologous to RNase H and shows endonuclease activity [9]. Mice have three PIWI-clade argonaute proteins; MIWI (PIWIL1), MILI (PIWIL2) and MIWI2 (PIWIL4) [10]. MIWI and MILI are cytosolic proteins and have piRNA-guided endoribonuclease or slicer activity, while MIWI2 is a nuclear protein and is proposed to recruit the histone or DNA methylation machinery to target genomic loci for transcriptional repression [11].

Although the proteins and/or enzymes involved in piRNA biogenesis are different between species, they have been conserved in the animal kingdom, and biogenesis occurs through almost the same mechanisms [12]. In mice, piRNAs are generated in two different ways, called primary and secondary (ping-pong cycle) pathways. The primary pathway may occur in the germline and somatic cells, and in this pathway, "long, single-stranded transcripts" are first generated from small RNA generative loci, piRNA clusters, and exported to the cytoplasmic granules termed "nuages", where several piRNA pathway components reside (Fig 1A) [10]. There are two different nuage types: the first consists of granules that contain MILI and its binding partner tudor protein 1 (TDRD1). These granules are also known as intramitochondrial cement-like granules or processing bodies (pi-bodies). The second one consists of MIWI2 and its binding partner TDRD9-containing granules, which are piP bodies [13]. TUDOR proteins were reported to have a selective role in correct binding of piRNAs to PIWI proteins [14]. Furthermore, a conserved protein called maelstrom (MAEL) is a nuage component and is essential for primary piRNA biogenesis, transposon silencing, and fertility in both flies and mice. The nucleocytoplasmic shuttling protein MAEL can bind RNA substrates such as transposons or piRNA precursors and deliver them to cytoplasmic nuages [15].

It has been suggested that long piRNA precursors created at the initial stage of the primary piRNA pathway are fragmented by the endonuclease activity of the mitochondrial protein PLD6 (MITOPLD), generating the 5' ends of piRNAs ("pre-piRNAs"). Furthermore, PLD6 is involved in the formation of nuages and has phospholipase activity, which hydrolyzes cardiolipin to generate phosphatidic acid and plays a role in the regulation of mitochondrial

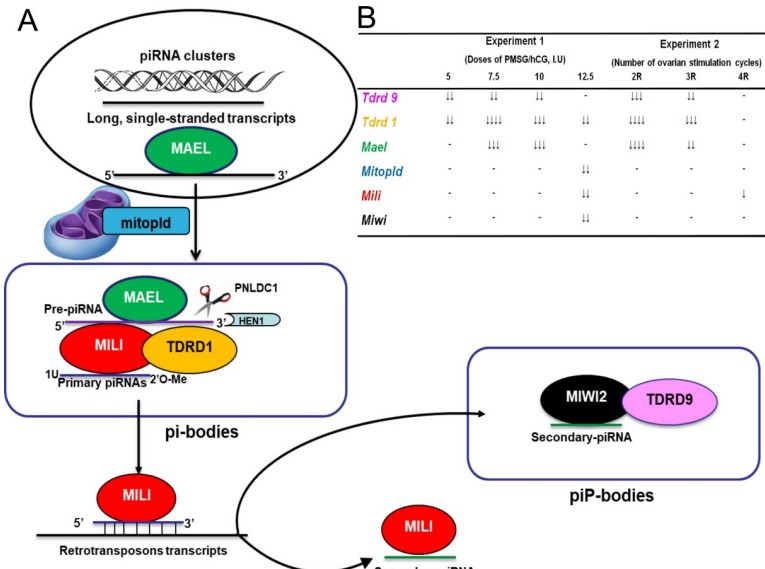

| | Experiment 1 (Doses of PMSG/hCG, I.U) | | | | Experiment 2 (Number of ovarian stimulation cycles) | | |
|---|---|---|---|---|---|---|---|
| | 5 | 7.5 | 10 | 12.5 | 2R | 3R | 4R |
| *Tdrd 9* | ↓↓ | ↓↓ | ↓↓ | - | ↓↓↓ | ↓↓ | - |
| *Tdrd 1* | ↓↓ | ↓↓↓↓ | ↓↓↓ | ↓↓ | ↓↓↓↓ | ↓↓↓ | - |
| *Mael* | - | ↓↓↓ | ↓↓↓ | - | ↓↓↓↓ | ↓↓ | - |
| *Mitopld* | - | - | - | ↓↓ | - | - | - |
| *Mili* | - | - | - | ↓↓ | - | - | ↓ |
| *Miwi* | - | - | - | ↓↓ | - | - | - |

**Fig 1. Schematic representation of the role of TDRD9, TDRD1, MAELLE, MITOPLD, MILI and MIWI proteins in the piRNA biogenesis and the effect of the ovarian stimulation on the gene expression.** (A) Schematic illustration of the role of investigated proteins in the primary and secondary (ping-pong cycle) piRNA biogenesis. In the primary pathway, long, single-stranded transcripts (black) produced from piRNA clusters are processed by the endonuclease activity of mitochondrial protein MITOPLD, generating the pre-piRNAs (purple) with uridine at the first nucleotide position (1U) and are loaded onto MILI, a Nuage protein [10, 13, 16]. At these stages, the nucleocytoplasmic shuttling protein MAEL binds piRNA precursors and delivers them to pi-bodies that contain MILI and its binding partner TDRD1 [13, 15]. The 3' end of pre- piRNAs is trimmed by a nuclease, PNLDC1, and modified by the methyltransferase HEN1 to generate mature primary piRNAs (blue). The mature primary piRNAs guide MILI to slice their complementary target RNAs such as retrotransposons transcripts and these processes cause the production of a series of secondary piRNAs (green) that associate with MILI and also MIWI2 (ping-pong cycle) [13,16]. MIWI2 and its interaction partner TDRD9 are found in piP bodies and don't have a slicer activity, but the MIWI2–piRNA complexes can regulate gene transcription by epigenetic mechanisms including histone modification and DNA methylation of target genes [12, 13]. (B) Effects of OS on the expression of *Tdrd9*, *Tdrd1*, *Mael*, *Mitopld*, *Mili*, and *Miwi*. 1–2 (↓), 2–4 (↓↓), 4–8 (↓↓↓), and 8–15 (↓↓↓↓) times significantly decrease in the expression levels of studied genes compared to control.

morphology [16]. The generated "pre-piRNAs" from small RNA generative loci bound to a PIWI protein (MILI), where it matures as a piRNA after the cleavage of the 3' end of pre-piR-NAs by a single-stranded-RNA exonuclease (PNLDC1 enzyme) and then 2′-O-Me modification at its 3′-ends by a methyltransferase (Hen1/HENMT1) [13]. The MILI-primary piRNA complexes slice retrotransposon-derived RNAs or target RNAs according to the guide sequence of primary piRNAs and produce both MILI-bound and MIWI2-bound secondary piRNAs (Fig 1A) [16]. These secondary piRNAs are generally derived from retrotransposons, and such repeated piRNA production is named the ping-pong cycle [17]. The ping pong cycle occurs mainly between MILI and MIWI2, where MILI slices the target RNAs via primary piR-NAs to guide the generation of piRNAs that bind to MIWI2. Binding of the secondary piRNA to MIWI2 provides the silencing of transposon repeats by direct DNA methylation [12].

In summary, the proteins involved in the biogenesis and function of piRNAs are critical for the biogenesis and/or correct functioning of piRNAs [18]. Defects in the function or expression of these proteins can affect the function or production of piRNAs. Consistent with this hypothesis, many studies have revealed that defective expression of the genes in piRNA bio-genesis can result in infertility, gametogenic dysfunction and increased accumulation of retro-transposon transcripts [19–21].

Some studies have reported that the piRNA pathway is regulated by sex hormones such as estrogen and testosterone [22, 23]. Estrogen was shown to downregulate the expression of *Miwi* and *Mili*. Moreover, many studies have demonstrated that exogenous gonadotropin treatment can increase the estradiol ($E_2$) levels [24, 25]. Therefore, OS may have an effect on the piRNA pathway. However, no study has examined the effect of exogenous gonadotropin treatment on the piRNA pathway, which is vital for genome stability. Thus, in the present study, we examined whether different doses of pregnant mare serum gonadotrophin (PMSG)/ human chorionic gonadotrophin (hCG) and repeated OS affect the expression of the *Mili*, *Miwi*, *Mael*, *Tdrd1*, *Tdrd9*, and *Mitopld* genes in mouse M2 (metaphase II) oocytes and ovarian tissue. In addition, we investigated the effect of OS on plasma $E_2$ levels and whether there is a relationship between $E_2$ levels and the expression of the genes involved in the piRNA pathway.

## Materials and methods

### Animals

All experimental protocols were performed according to the guidelines for the ethical procedures of experimental animals and were approved by the local animal ethics committee of Cumhuriyet University, Sivas, Turkey (approval No: 29.12.2015–92). Female BALB/c mice (n = 56) were used for the experimental procedures at the age of 8–10 weeks. The mice were kept under a 12-h light/dark cycle with a temperature of 20–22°C and relative humidity of 50–65% and had free access to standard laboratory food and water.

### Study groups and gonadotropin stimulation

Our study was divided into two experimental stages to investigate the effects of both different doses of exogenous gonadotropin (experiment 1) and repeated OS (experiment 2) on the expression of some proteins involved in the PIWI-piRNA pathway. Mice in experiment 1 were injected with 5 (group I), 7.5 (group II), 10 (group III) and 12.5 I.U. (group IV) PMSG (Sigma, St Louis, MO, USA), followed 48 hours later by 5, 7.5, 10, and 12.5 I.U. hCG (Sigma, St Louis, MO, USA) intraperitoneally (i.p.) in each group, respectively [26,27]. Hormones were administered i.p. in 0.1 mL of 0.9% sterile NaCl solution. Controls were injected twice with 0.1 mL sterile serum physiologic isotonic solution. The mice in experiment 2 were injected i.p. with 5 I.U. PMSG and 48 hours later with 5 I.U. hCG, and 2 to 4 rounds (2R, 3R and 4R groups) of stimulations were performed with intervals of 1 week between each. Each group contained seven mice, and daily vaginal smears were taken from the female mice to establish the estrous cycle.

### Collection of blood samples, MII oocytes and ovarian tissues

Mice from each group were anesthetized with an i.p. injection of diazepam (5 mg/kg, Stesohid; Dumex, Copenhagen, Denmark) and ketamine (200 mg/kg, Sante Animale, Brussels, Belgium). Blood samples were collected from the left ventricle of the heart in a heparinized syringe. Plasma samples were obtained by centrifuging the blood samples at 3000 rpm for 10 min and then were stored at -80°C until $E_2$ assays by the enzyme linked immunosorbent assay (ELISA) method. After the collection of blood samples, the mice from all the study groups were euthanized by i.p. injection of sodium pentothal (200 mg/kg). MII oocytes (at least 20) were collected from the oviducts of superovulated mice 16 hour after an hCG injection and were placed in RNAlater buffer (Qiagen, Hilden, Germany). Moreover, the ovarian tissues were taken from all the groups in experiments 1 and 2. One ovary from each animal was stored in 10% buffered neutral formalin for immunofluorescence analysis, and the other was stored

in RNAlater buffer for quantitative real-time polymerase chain reaction (qPCR) studies. The tissues and oocytes in RNAlater buffer were stored at −80˚C before RNA extraction.

## Measurement of 17-β-estradiol

The plasma $E_2$ levels were measured with an ELISA kit (17-β-estradiol mice, Elabscience) according to the instructions of the manufacturers.

## RNA extraction and cDNA synthesis

Total RNAs were isolated from MII oocytes and ovarian tissues of all the groups by using an RNeasy Plus Micro Kit and RNeasy Mini Kit (Qiagen, Hilden, Germany), respectively, according to the instructions of the manufacturers. The concentration and quality of the RNA samples were determined by using a microplate spectrophotometer (Epoch, Biotek, USA), and then, equivalent quantities of RNA from each sample were reverse-transcribed to cDNA using an iScript cDNA synthesis kit (Bio-Rad, Hercules, CA, USA) according to the manufacturer's recommendation.

## qPCR for *Mili*, *Miwi*, *Mael*, *Tdrd1*, *Tdrd9* and *Mitopld*

qPCR was carried out on a Rotor Gene 6000 real-time PCR instrument (Qiagen, Doncaster, Australia) using Luna Universal qPCR Master Mix (New England BioLabs®, Inc.) according to the manufacturer's specifications. For amplification, 2 µl of diluted cDNA was added to each real-time qPCR mixture, containing 10 µl of Luna Universal qPCR Master mix and 1 µM of the target gene-specific forward and reverse primers in a final volume of 20 µl. The primer sequences used in amplification are shown in Table 1. qPCR consisted of 95˚C for 1 min to activate the polymerase, followed by 40 cycles at 95˚C for 15 s and 60˚C for 30 s. Mouse *Gapdh* was used as a housekeeping gene control. The specificity of the qPCR products was confirmed by analysis of melting curves. The relative gene expression data were analyzed using the 2-ΔΔCt method. The samples were analyzed in triplicate, and cycle threshold (Ct) values greater than 35 were excluded from the statistical evaluation.

## Immunofluorescence labeling

MILI, MIWI, MAEL, TDRD1, TDRD9 and MITOPLD proteins in the ovarian tissues were determined using immunofluorescence staining. Ovarian tissues of all the study groups were fixed in 10% buffered neutral formalin for 30–36 hours. After dehydration, clearing and paraffin embedding, the tissues were cut by a microtome into 5-µm thick sections and placed on poly-L-lysine-coated slide glasses. Following deparaffinization in xylene and hydration through a series of graded ethanol, the sections were washed in distilled water. After the sections were heated in 10 mM Tris/ 1 mM EDTA buffer, pH = 9, (anti-TDRD9, anti-TDRD1, and anti-MILI staining) or EDTA buffer, pH = 8, (anti-MAEL, anti-MITOPLD, and anti-MIWI staining) two times for 30 min at maximum power and for 10 min at medium power, respectively, in a domestic microwave oven, they were cooled to RT and rinsed three times in phosphate-buffered saline with Tween-20 (PBS-Tween-20). For inhibition of endogenous peroxidase activity, the sections were treated with 3% hydrogen peroxide in PBS for 30 min at RT and then washed 3 times with PBS-Tween-20. Later, the sections were blocked with blocking solution with 5% normal goat serum, 1% BSA, 0.1% Triton X-100, and 0.05% Tween 20 in PBS for 45 min at RT. After blocking, the sections were incubated overnight at 4 ºC with primary antibodies against mouse TDRD9 (ab118427, Abcam, 20 µg/mL), TDRD1 (739206, Invitrogen, 20 µg/mL), MAEL (sc-398925, Santa Cruz 1/100), MITOPLD (CAC07666, Biomatik, 1/50),

**Table 1. List of primers used for qRT-PCR.**

| Gene[a] | | Primer sequence (5′ → 3′) |
|---|---|---|
| *Mili (Piwil 2)* | FW | CAGAAGTGTTTTGAAGCCTTTGATA |
| | RV | TGGTGCTGATTTTCTTCTGAACTAC |
| *Miwi (Piwil 1)* | FW | Optimized and specific PCR primers were used (Qiagen QuantiTect primer assays; Cat no: QT00130473) |
| | RV | |
| *Mael (Maelstrom)* | FW | GAAGCTAAGAGTTGAGAGTCCAGGAT |
| | RV | GATGCTCTCTAGTAAGCGGGTAATTC |
| *Tdrd1* | FW | TTGAATCAGTCCTTAGCAGACTACTGT |
| | RV | ACTTGGTAAGATCTCCTTGACTAGAGC |
| *Tdrd9* | FW | TCCAGTGTGACTTTAGAAGAACAGAA |
| | RV | ACATATTTGACATCAGGAACTGTGAC |
| *Mitopld (Pld6)* | FW | CTAGGCTACATGCACCACAAGTT |
| | RV | ACATACTCGGTGTCCTCCATAATC |
| *Gapdh* | FW | Optimized and specific PCR primers were used (Qiagen QuantiTect primer assays; Cat no: PPM02946E) |
| | RV | |

FW, forward primer; RV, reverse primer

MILI (MBS8245645, MyBioSource, 1/100) and MIWI (MBS 8245645, MyBioSource, 1/100). Then, the sections were washed 3 times with PBS-Tween-20 for 5 min each time and incubated with FITC-conjugated anti-rabbit (sc-2012, Santa Cruz) and anti-mouse (sc-2010, Santa Cruz) secondary antibodies (dilution 1:250) for 1 hour at RT. Last, these sections were washed again 3 times with PBS for 5 min each, and then, nuclear staining was performed with 0.5 μg/mL DAPI (4', 6'-diamidino-2-phenylindole dihydrochloride, Fluka, USA) for 10 min at RT and evaluated on a fluorescence microscope (Olympus BX51, Tokyo, Japan). At least three different sections of each sample were analyzed. The tissue sections without the primary antibody served as the negative control for each staining. The sections were visualized and photographed with a fluorescence microscope (BX-51 Olympus, Japan).

## Image analysis

Quantification of immunofluorescence staining in the ovary was performed by ImageJ software (National Institute of Mental Health, Bethesda, Maryland, USA). The intensity settings were constant for all sections. The ovarian sections from each animal were evaluated. The relative fluorescence for each protein was quantified as previously described [28–30]. Briefly, after channel separation (RGB) of color images, each protein labeled in the green channel was quantified for the mean pixel intensity. The selected area (region of interest) was measured at least from five randomly selected regions (in square pixels were equal) of each section. Values from all analyses were entered into one-way analysis of variance (ANOVA), followed by pairwise comparisons in Tukey's HSD test.

## Statistical analysis

Statistical evaluation was performed using SPSS version 22 (SPSS, Inc., Chicago, IL, USA). The $E_2$ levels were compared between all groups by one-way ANOVA, and paired comparisons were performed with Tukey t tests. The qPCR data were analyzed with the ΔΔCt method, and ANOVA and Student's t tests were used as appropriate. The relative quantification of the gene expression levels was determined by the ΔΔCt method using the RT2 profiler RT-PCR Array

Data Analysis Programme (Qiagen, Inc., Valencia, CA, USA, version 3.5) with Student's t tests. The statistical significance level was set at $p \leq 0.05$.

## Results

### Plasma 17-β estradiol levels

The comparison of $E_2$ levels measured in all groups in experiments 1 and 2 is shown in Table 2. A significant difference was determined in terms of the $E_2$ levels between the groups in both experiments 1 and 2 (p = 0.0001; p<0.05). Paired comparisons with Tukey's test revealed a significant difference between the control group and groups 1, 2 and 3 in experiment 1 (p = 0.0080, 0.0001 and 0.0050, respectively). There was also a statistically significant difference between group 1 and group 4 (p = 0.045), but no significant differences were found in the paired comparison between the other groups in experiment 1 (p > 0.05). In experiment 2, a significant difference was found between the control and 2R groups (p = 0.0001). However, the difference between the control and 3R and 4R groups (p = 0.078 and 0.957, respectively) was not significant.

### The effects of OS on the expression of the *Mili*, *Miwi*, *Mael*, *Tdrd1*, *Tdrd9* and *Mitopld* genes

The effect of OS on the expression of *Mili*, *Miwi*, *Mael*, *Tdrd1*, *Tdrd9* and *Mitopld* was examined by qPCR, both in ovarian tissues and MII oocytes. Fig 2 shows the comparison of these genes in the ovarian tissues between the control group and the groups with different doses of exogenous gonadotropin (experiment 1) and repeated OS (experiment 2) in ovarian tissue.

In experiment 1, the mRNA expression of *Tdrd1* was 2.47-, 8.09-, and 5.30-fold and that of *Tdrd9* was 2.39-, 3.09-, 2.81-fold lower than that of the controls in groups 1, 2, and 3, respectively. The ovarian *Mael* mRNA expression of mice in experiment 1 had a 5.55- and 4.75-fold decrease in groups 2 and 3, respectively (p<0.05). However, the decreases in *Tdrd1* and *Tdrd9* expression in group 4 and in *Mael* expression in groups 1 and 4 were not significant. In addition, in experiment 1, the highest decrease in the expression of *Tdrd1*, *Tdrd9* and *Mael* was clearly observed in group 2 (3.09-, 8.09-, and 5.55-fold decreases, respectively). Furthermore, when the expression of *Mitopld*, *Mili* and *Miwi* was evaluated according to the groups in experiment 1, a significant decrease was found only in group 4 (2.44-, 2.80-, and 2.10-fold decreases, respectively).

In experiment 2, the *Tdrd1*, *Tdrd9*, and *Mael* mRNA levels were 4.29-, 14.34- and 11.63-fold lower in the 2R group and 4.59-, 3.72-, and 2.91-fold lower in the 3R group than in the control group, respectively. Although the expression levels of these three genes in the 4R group decreased slightly, these changes were not statistically significant. The *Tdrd1*, *Tdrd9* and *Mael* expression levels showed the strongest decreases in 2R among all the groups in experiment 1 and experiment 2. Moreover, the relative expression levels of the *Mili* genes decreased significantly only in the 4R group (-1.45-fold). However, no significant differences were found in the expression levels of *Mitopld* and *Miwi* between the control group and the groups in experiment 2.

Finally, when the expression levels of these 6 genes were examined in MII oocytes, we found they were expressed at relatively low levels compared to those in the ovarian tissue. Exogenous gonadotropin administration further reduced the expression levels that were already low, and the reduction rates were consistent with the gene expression data of the ovarian tissues in terms of the groups (data not shown). However, since the Ct values of these genes in M2 oocytes were generally 30 and above, they could not be statistically evaluated.

Table 2. Plasma E2 levels in the groups in experiments 1 and 2.

| | Control (n = 7) | Experiment 1 | | | | Experiment 2 | | | p |
|---|---|---|---|---|---|---|---|---|---|
| | | Group1 (n = 7) | Group2 (n = 7) | Group3 (n = 7) | Group4 (n = 7) | 2R (n = 7) | 3R (n = 7) | 4R (n = 7) | |
| E2 (pg/mL;X±S) | 92.9±14 | 134.3±15 | 147.8±20 | 135.0±21 | 115.0±27 | 160.3±20 | 119.3±21 | 98.2±15 | 0.0001* |

Data are expressed as the mean ±standard deviation. E2, 17-β estradiol.

*p<0.05

### Effect of OS on the immunostaining intensity of MILI, MIWI, MAEL, TDRD1, TDRD9 and MITOPLD in the ovarian tissues

The expression of the MILI, MIWI, MAEL, TDRD1, TDRD9 and MITOPLD proteins was investigated in mouse ovarian tissues and visualized by fluorescence microscopy. As shown in Fig 3A–3L, all six proteins were expressed in the interstitial stromal cells, which are distributed in the spaces between follicles, and in the medulla region of the ovary, but no expression was observed in the granulosa cells and oocytes. These proteins were expressed mostly in the theca internal layers of growing preantral and antral follicles (red arrow) and stromal cells (white arrow) surrounding the follicles.

ImageJ analysis showed that in experiment 1, there was a significant decrease in the TDRD1 and TDRD 9 protein levels in groups 1, 2, and 3. In experiment 2, TDRD1 expression decreased significantly in the 2R group as did TDRD 9 expression in the 2R and 3R groups compared to the control group (Fig 4A and 4B). Furthermore, the level of MAEL expression in groups 2, 3, and 4 in experiment 1 and group 2R in experiment 2 was significantly lower than that in the control group, but the decrease observed in groups 1, 3R and 4R was not significant (Fig 4C). Similar to the qPCR results, the lowest levels of TDRD1, TDRD 9 and MAEL for experiment 1 and experiment 2 were detected in groups 2 and 2R, respectively. Although there was a reduction in the expression levels of MITOPLD in all experimental groups, only the decrease in group 4 was significant (Fig 4D). There was a mild decrease in MILI expression in groups 4 and 4R compared to the control group, but this change did not reach statistical significance (Fig 4E). Finally, the MIWI expression levels were similar in all groups (Fig 4F).

## Discussion

In this study, the effect of different doses of PMSG and hCG treatment and repeated OS on the expression levels of the *Mili*, *Miwi*, *Mael*, *Tdrd1*, *Tdrd9*, and *Mitopld* genes, which are involved

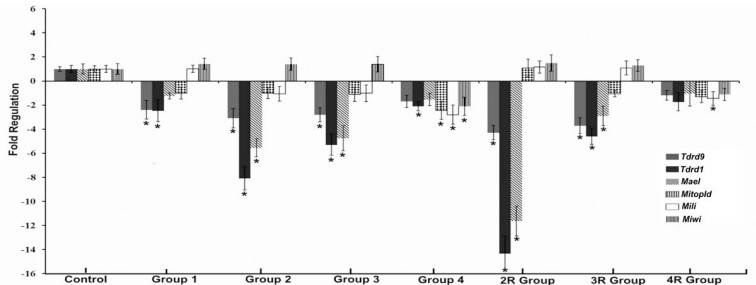

**Fig 2. Effects of ovarian stimulation on *Tdrd9*, *Tdrd1*, *Mael*, *Mitopld*, *Mili*, and *Miwi* mRNA levels in ovarian tissue of the study groups.** Fold-Regulation (FR) represents fold-change results in a biologically meaningful way. Fold-change values greater than one indicates a positive- or an up-regulation, and the FR is equal to the fold-change. Fold-change values less than one indicate a negative or down-regulation, and the FR is the negative inverse of the fold-change. *p<0.05.

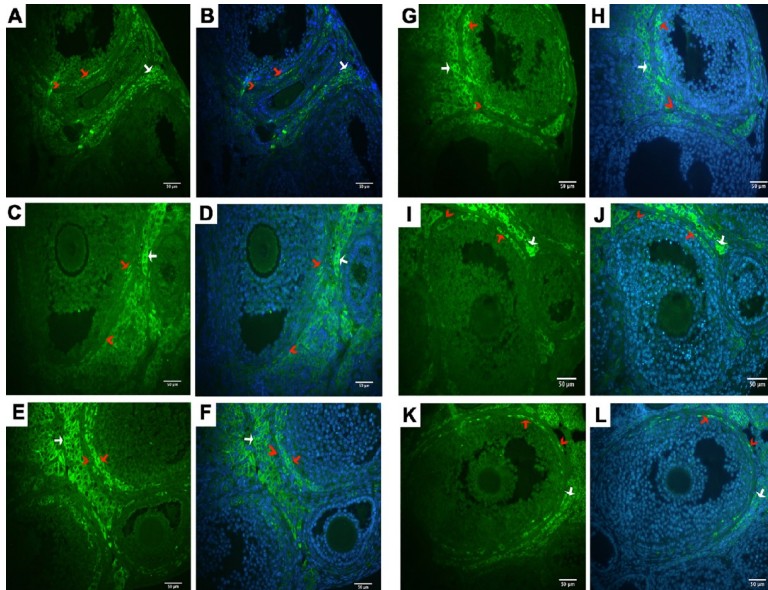

**Fig 3. Immunofluorescence analysis of TDRD9, TDRD1, MAEL, MITOPLD, MILI, and MIWI in the ovarian tissue sections of the control group.** Representative immunofluorescence images of TDRD9 (A-B), TDRD1 (C-D), MAEL (E-F), MITOPLD (G-H), MILI (I-J), and MIWI (K-L) (400X magnification). Red arrow, theca interna; Red arrow heads, theca externa; white arrow, stromal cells. MILI, MIWI, MAEL, TDRD1, TDRD9 and MITOPLD stained with FITC. Blue = DAPI nuclear counterstain, green = FITC.

in biogenesis and function of piRNAs were examined for the first time. Furthermore, the effect of controlled OS on the plasma $E_2$ levels and the relationship between the $E_2$ levels and the expression of these genes was also investigated.

Several previous studies showed that the stimulation of ovaries with exogenous gonadotropins led to high levels of $E_2$ secretion [25, 31]. Our results demonstrated that although increases were observed in the $E_2$ levels in all groups both in experiment 1 and in experiment 2, the levels increased significantly after treatment with 5, 7.5 and 10 I.U. PMSG/hCG and two repeated rounds of OS. We observed that the highest reduction rate in $E_2$ levels was in groups 2R and 2, respectively, but the difference between groups 1, 2 and 3 was not significant (Table 1).

In previous studies, it was found that high levels of $E_2$ decreased the quality of the oocytes and embryos [32, 33] and prevented implantation of the fertilized ovum through negative effects on the endometrium [31]. Given the negative effects of high $E_2$ levels on reproductive biology, exogenous gonadotropin doses, which are widely used in assisted reproductive techniques, should be evaluated and optimized.

In our study, when the effects of OS on the expression levels of the genes involved in the piRNA pathway were evaluated by qPCR in ovarian tissues depending on both the dose of exogenous gonadotropin administration and the number of OS cycles performed, we found that the *Tdrd1*, *Tdrd9*, and *Mael* expression levels decreased in all groups. However, the decreases observed in groups 2, 3, 2R, and 3R for all 3 genes were significant (Fig 2). Furthermore, the immunostaining results of these proteins were broadly consistent with the qPCR findings. According to the results of both qPCR and immunofluorescence analysis, the highest decreases in the expression of *Tdrd9*, *Tdrd1*, and *Mael* were found in 7.5 I.U. PMSG/hCG treatment group among the dose-dependent groups and in the 2R group among the repeated OS groups (Figs 2 and 4A-4C and S1–S6 Figs). Moreover, the highest reduction of mRNA and

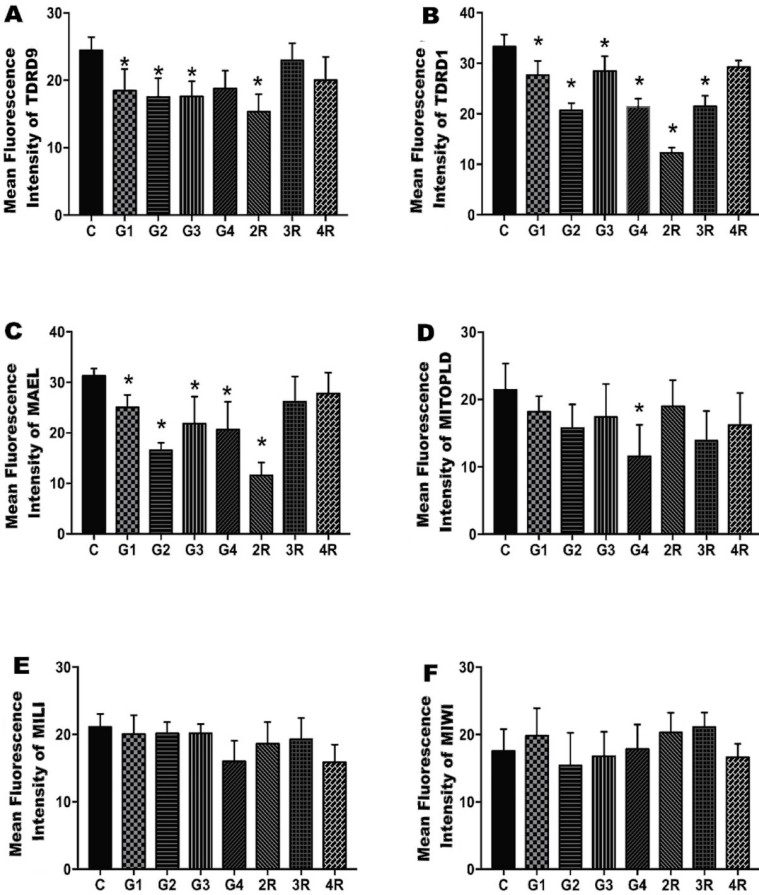

**Fig 4. The intensity of immunoexpression of TDRD9, TDRD1, MAEL, MITOPLD, MILI, and MIWI in the ovarian tissue sections of the study groups.** Comparison of the expression of TDRD9 (A), TDRD1 (B), MAEL (C), MITOPLD (D), MILI (E), and MIWI (F) in the ovarian tissue between the control and the study groups. Data are presented as the mean ± SD. C, Control; G, Group. $^*p<0.05$.

protein expression of these three genes was found in group 2R among all the study groups. In parallel with these results, the observation that the increase in plasma $E_2$ levels was the highest in group 2R, then next highest in group 2 and the lowest in groups 4 and 4R, suggested that the expression levels of these genes may be associated with plasma E2 levels. In a recent study, $E_2$ injection was shown to suppress the transcription levels of *Miwi* and *Mili* in the ovary of female Dabry's sturgeon [34]. Furthermore, Wang et al. demonstrated that hCG and $E_2$ suppressed the expression of *Mili* in ovarian tissue *in vivo* and *in vitro* [23]. In another study, it was found that administration of hCG and luteinizing hormone-releasing hormone reduced *Mili* and *Miwi* expression at both the protein and mRNA levels in the ovaries of *Odontobutis potamophila* [35]. These findings and the data of our study indicated that the expression of the genes involved in the piRNA pathway may be regulated by both estradiol and gonadotropins. These hormones play an important role in germline development and embryogenesis via their impacts on the piRNA pathway, in addition to their known effects on the reproductive system.

In some reports, the proteins examined in our study were shown to be expressed at a moderate level in ovarian tissue [35–38]. In our study, as a result of the immunofluorescence analysis of ovarian tissues, we observed that the expression of these proteins was generally low (Figs 3A–3L and 4, S1–S6 Figs). Furthermore, we found that the MILI, MIWI, MAEL, TDRD1,

TDRD9, and MITOPLD proteins were expressed in interstitial stromal cells, which are distributed in the spaces between follicles and in the medulla region of the ovary, but no expression was observed in the granulosa cells and oocytes (Fig 3A–3L). According to the two-cell/two-gonadotropin theory, granulosa cells are special cells in which estrogen synthesis takes place [39]. The observation that the proteins examined in our study were not expressed in these cells supports the idea that the expression of these proteins was regulated by $E_2$ levels.

In the present study, when changes in the expression of the *Mitopld*, *Mili*, and *Miwi* genes were examined in ovarian tissues, we found that all 3 genes significantly decreased only in the 10 I.U. PMSG/hCG treatment group among the groups treated with different doses of PMSG/hCG, and the immunohistochemical analysis results of these 3 proteins were largely similar to the gene expression data. Furthermore, in the groups receiving repeated injections of PMSG/hCG, only the *Mili* gene significantly decreased in group 4R. All these findings suggest that the effect of plasma $E_2$ levels on the expression of these proteins may vary from protein to protein in the piRNA pathway or there may be a complex regulatory pattern between these proteins. We found that the expression levels of all the genes we examined were the least affected in the 5 I.U. PMSG/hCG treatment group among the study groups ($p > 0.05$). However, collectively, the administration of exogenous gonadotropin led to substantial changes in the expression of these genes that play a crucial role in piRNA functions and biogenesis (Fig 1A and 1B). piR-NAs have strong effects on the occurrence of embryo implantation and they mediate the formation of healthy gametogenesis, especially in terms of genome integrity [5]. In a study, the purpose of which was to analyze the sncRNA expression profile of the spent culture media after fertilization and to investigate the relationship between the sncRNA and morphofunctional properties of gametes and the resulting embryos, it was revealed that some piRNAs have a significant effect on the occurrence of implantation, oocyte-cumulus complex and two pronuclei embryo numbers [40]. In other studies, it has been shown that abnormal conditions that cause defects in piRNA functions can lead to some important negative effects on the reproductive system, such as increased embryonic lethality [41] and infertility [19]. Furthermore, a number of piRNA pathway proteins are essential for the function and biogenesis of piRNAs and retrotransposon silencing [17]. Therefore, potential defects in the functions and/or expression of these proteins can affect the activity or levels of piRNAs, and such situations may cause genome instability, inducing failure in gametogenesis and infertility. Kabayama et al. found that when the expression of *Mitopld*, *Mili*, and *Miwi* was depleted by gene knockout, it resulted in a significant decrease in piRNA levels compared to those of the wild type, accompanied by an increase in retrotransposon transcripts [16]. Sienski et al. demonstrated impaired transposon silencing and female fertility in mutant flies lacking the HBG box domain of the MAEL protein [42]. Furthermore, Castaneda et al. found that the piRNA levels in *Mael* mutant mice showed an approximately 10-fold reduction [43]. It was reported that loss-of-function mutations in another piRNA pathway protein, MITOPLD, which is involved in the formation of nuages and has endonuclease activity, caused significant defects in piRNA biogenesis and spermatogenesis in mice [36,44]. Based on these results, we concluded that significant reductions in the expression of these genes in our study groups, especially in groups 2 and 2R and groups 4 and 4R, may lead to changes in piRNA biogenesis and the function of piRNAs. Such a situation may lead to some risks for genome integrity and chromosome number stability and consequently may have negative effects on the perfect occurrence of gametogenesis. In many studies, it was reported that in *Miwi*, *Mili*, *Mitopld*, and *Tudor* mutant and/or knockout mice, there were defects in germ cell formation, some of which cause infertility [14, 45–48]. Therefore, these results are consistent with our last hypothesis.

## Conclusions

In conclusion, it was observed that OS performed with exogenous gonadotropins significantly reduced the expression of the *Mili*, *Miwi*, *Mael*, *Tdrd1*, *Tdrd9*, and *Mitopld* genes in some groups. These reduction rates were the highest in the group 2R and then 7.5 I.U. PMSG/hCG treatment group in terms of *Tdrd1*, *Tdrd9* and *Mael* expression both at the protein and mRNA levels. The fact that the highest increase in plasma $E_2$ levels was found in these groups, and these proteins were not observed in granulosa cells where estrogen synthesis occurs, indicates that changes in the expression levels of these 3 genes may be related to $E_2$ levels. When the effects of proteins involved in biogenesis and functions of piRNA on gametogenesis, embryogenesis and maintenance of genome integrity are considered, exogenous gonadotropin administration may increase the risk of genetic instability depending on the dose and the number of repetitions. Further studies are needed to examine the effects of exogenous gonadotropin administration on this pathway to help prevent the potential detrimental effect of OS, which is also used in IVF, especially on genome integrity and gametogenesis.

## Supporting information

**S1 Fig. Immunolocalization of the TDRD9 protein in the ovarian tissue sections of the control and study groups.** The expression and distribution of TDRD9 (stained with FITC, green) in the control (a), group 1 (b), group 2 (c), group 3 (d), group 4 (e), group 2R (f), group 3R (g), and group 4R (h) were evaluated by immunofluorescence staining (200X magnification). Green = FITC.
(TIF)

**S2 Fig. Immunolocalization of the TDRD1 protein in the ovarian tissue sections of control and study groups.** The expression and distribution of TDRD1 (stained with FITC, green) in the control (a), group 1 (b), group 2 (c), group 3 (d), group 4 (e), group 2R (f), group 3R (g), and group 4R (h) were evaluated by immunofluorescence staining (200X magnification). Green = FITC.
(TIF)

**S3 Fig. Immunolocalization of the MAEL protein in the ovarian tissue sections of the control and study groups.** The expression and distribution of MAEL (stained with FITC, green) in the control (a), group 1 (b), group 2 (c), group 3 (d), group 4 (e), group 2R (f), group 3R (g), and group 4R (h) were evaluated by immunofluorescence staining (200X magnification). Green = FITC.
(TIF)

**S4 Fig. Immunolocalization of the MITOPLD protein in the ovarian tissue sections of the control and study groups.** The expression and distribution of MITOPLD (stained with FITC, green) in the control (a), group 1 (b), group 2 (c), group 3 (d), group 4 (e), group 2R (f), group 3R (g), and group 4R (h) were evaluated by immunofluorescence staining (200X magnification). Green = FITC.
(TIF)

**S5 Fig. Immunolocalization of the MILI protein in the ovarian tissue sections of the control and study groups.** The expression and distribution of MILI (stained with FITC, green) in the control (a), group 1 (b), group 2 (c), group 3 (d), group 4 (e), group 2R (f), group 3R (g), and group 4R (h) were evaluated by immunofluorescence staining (200X magnification). Green = FITC.
(TIF)

**S6 Fig. Immunolocalization of the MIWI protein in the ovarian tissue sections of the control and study groups.** The expression and distribution of MIWI (stained with FITC, green) in the control (a), group 1 (b), group 2 (c), group 3 (d), group 4 (e), group 2R (f), group 3R (g), and group 4R (h) were evaluated by immunofluorescence staining (200X magnification). Green = FITC.
(TIF)

## Author Contributions

**Investigation:** Ismail Sari, Erkan Gumus.

**Methodology:** Ismail Sari, Erkan Gumus, Ahmet Sevki Taskiran.

**Project administration:** Ismail Sari.

**Supervision:** Lale Karakoc Sokmensuer.

**Validation:** Lale Karakoc Sokmensuer.

**Writing – original draft:** Ismail Sari.

**Writing – review & editing:** Ismail Sari, Erkan Gumus, Lale Karakoc Sokmensuer.

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
