## [Decision Letter · Decision Letter 0]

8 Jan 2020

PONE-D-19-34298

Effect of ovarian stimulation on the expression of piRNA pathway proteins

PLOS ONE

Dear Dr. Sari,

Thank you for submitting your manuscript to PLOS ONE. After careful consideration, we feel that it has merit but does not fully meet PLOS ONE’s publication criteria as it currently stands. Therefore, we invite you to submit a revised version of the manuscript that addresses the points raised during the review process.

Rationale of the work, experimental design, choice of doses need to be explained well. The analyses and significant data obtained are limited to gene and protein expression related to piRNA. Hence author should add additional experiments or discuss more about the relevant publications to co-relate the study results to support the conclusion specifically.

We would appreciate receiving your revised manuscript by Feb 22 2020 11:59PM. To enhance the reproducibility of your results, we recommend that if applicable you deposit your laboratory protocols in protocols.io, where a protocol can be assigned its own identifier (DOI) such that it can be cited independently in the future. For instructions see: http://journals.plos.org/plosone/s/submission-guidelines#loc-laboratory-protocols

We look forward to receiving your revised manuscript.

Kind regards,

Academic Editor

PLOS ONE

Additional Editor Comments:

Rationale of the work, experimental design, choice of doses need to be explained well. The analyses and significant data obtained are limited to gene and protein expression related to piRNA. Hence author should add additional experiments or discuss more about the relevant publications to co-relate the study results to support the conclusion specifically.

Journal Requirements:

Reviewers' comments:

Reviewer's Responses to Questions

**Comments to the Author**

1. Is the manuscript technically sound, and do the data support the conclusions?

Reviewer #1: Partly

Reviewer #2: No

Reviewer #3: Partly

Reviewer #4: Partly

Reviewer #5: Partly

2. Has the statistical analysis been performed appropriately and rigorously? 

Reviewer #1: Yes

Reviewer #2: I Don't Know

Reviewer #3: Yes

Reviewer #4: Yes

Reviewer #5: I Don't Know

3. Have the authors made all data underlying the findings in their manuscript fully available?

Reviewer #1: Yes

Reviewer #2: Yes

Reviewer #3: No

Reviewer #4: Yes

Reviewer #5: No

4. Is the manuscript presented in an intelligible fashion and written in standard English?

Reviewer #1: Yes

Reviewer #2: No

Reviewer #3: Yes

Reviewer #4: No

Reviewer #5: Yes

5. Review Comments to the Author

Reviewer #1: The manuscript is about the influence of ovarian stimulation on the genes that are involved in piRNA biogenesis. Though their findings are interesting, manuscript lacks clear and stronger experiments that are required in various parts of the manuscript making the manuscript technically flawed.

1. As reproductive cycle and age are correlated, why mice of 8-10 weeks were chosen for the study is not explained anywhere in the manuscript.

2. What’s the reason for dose and time period for hCG and PMSG? Dose and intervals might have induced detrimental effects. There is no strong basis or explanation provided from the authors side in this context.

3. Why FSH and LH levels , body weights were not measured as a control for ovarian stimulation occurance.

4. The exact function of Mili, Miwi, MaeI, Tdrd1, Tdrd9 , Mitopld genes in the piRNA pathway is not very well explained in the manuscript.

5. The authors should consider making an illustration or model figure for piRNA pathway and corresponding changes upon ovarian stimulation.

6. The authors should present qPCR results in the form of a bar /column chart.

7. The manuscript carries typos and grammatical errors at various places which has to be keenly addressed For eg. in abstract line 9 “ remarkable decreases”, Page 2 line 10 “piRANA”, Page 9 , Page 19 “was found that” and reference 6 “ editor” etc. Hence it has to be proof read.

8. The images are not very clear, hence a zoomed portion in the tissue section has to be included.

9. Image analysis can be well explained separately in the methods.

10. It has been said in the conclusion that genome instability occurs on external gonadotrophin administration depending on the dose and the repetition of administration. Did author test on any genome instability that arises out of disrupted piRNA biogenesis (for eg microsatellite) detection like using PCR were done on the tissues ?

11. It has been found that piRNA has different roles in cytoplasm and nucleus. The author has to experimentally or theoretically address pathway in which compartment Mili, Miwi, MaeI, Tdrd1, Tdrd9 , Mitopld genes are specifically affected.

12. The molecular mechanisms underlying piRNA biogenesis and functions are complex and diverse,the paper must provide more emphasis on step by step piRNA biogenesis and overall biological impact of the study.

Reviewer #2: This manuscripts studies effect of PMSG/hCG and ovarian stimulation on the piRNA pathway. Despite the interesting observations, the manuscript lacks detailed reasoning for the design of the experiments. Additionally, the manuscript should be carefully proofread for grammatical errors and typos which are present throughout the manuscript.

1. How were dose ranges for PMSG and hCG decided? Would the authors see difference in protein expression levels in other dose concentrations?

2. Pictorial summarization/representation of the observed data pathway would help comprehend the data with ease.

3. Immunofluorescence data could be better represented by enlarged images.

4. Typos and grammatical errors are abundant. For eg. in abstract line 5, “ 7.5 PMSG/hCG” – IU is missing, page 2 line 4, “ embryo and offspring elicited by OS”, page 2 line 15 “transposons, regulation of some gene expression”, page 6 line 21, “investigated in our study examined in MII oocytes”, page 7 line 18 and 20 “ 1.u”, page 7 line 29 “caused to an increase”, page 9 line 19 “ Kabayama et al…..retrotransposon transcripts [36]”- please rephrase this sentence.

5. Please pay attention to Author contribution in page 13. Who is J.L?

Reviewer #3: In this manuscript, the authors have attempted to investigate the effect of exogenous gonadotropin injections and ovarian stimulation (OS) on the expression of genes/proteins involved in piRNA biogenesis. Given that OS is essential for infertility treatment, it is important to know the exact mechanisms responsible for these detrimental effects of OS in order to increase the success of IVF. The experimental design is logical but the analysis is limited to expression of proteins/genes related to piRNA. Based on the results presented and references cited, the expression of the proteins involved with piRNA may be regulated by both estradiol and gonadotropins. However, the authors have not shown the mechanism of this connection.

Below are some experimental suggestions and questions, which when addressed appropriately would make the manuscript commendable for publication.

Introduction and methods:

1. The phrase ‘piRNA pathway’ is not clearly defined, whether it refers to biogenesis of piRNA or functions hereafter. Also, piRNA and hormones should be elaborated with appropriate references.

2. What is the difference between ‘controlled OS’ and ‘OS’? it is not clear what was followed in the experimental setup as the terms are used interchangeably in the manuscript.

3. Immunofluorescence labelling in methods is not clear. “After the sections were heated in 10 mM……. phosphate-buffered saline with tween-20 (PBS-Tween-20)” – consider simplifying and rephrasing this sentence.

Results:

1. It is not clear why the authors compare the results of both the experiments 1 and 2 together only in the immunoexpression studies.

2. Table 3 results would be better if represented as a bar graph to highlight the differential expression levels.

3. The sentence “Based on this studies and the obtained data from the ……………………………………and these increasing rates were similar in groups 1, 2 and 3” needs clarification. The authors have not explained the relationship between the different hormone injections and their respective effect on receptors. Rather they have just made statements about hormone injections and receptors with cited references. As mentioned by the authors, the relationship between E2, FSH, LH and the receptor regulation of these hormones is complex and requires valid experimental evidence. The authors should include experimental data on the expression levels of these receptors under the current experimental conditions to make a relevant connection.

4. What is the quality of the oocyte and embryo developed under the tested experimental conditions. This result along with the data on FSH, LH, E2 hormone/receptor levels will help to connect the dots.

5. Why are expression levels of Tdrd 9, Tdrd1, and Mael decreased in 2R but not in 3R and 4R. There seems to be a positive correlation between the rounds of OS and the rescue in expression levels compared to the control. What is the reason for this?

6. Fig1 legend is not clear. What is control (a) and control (j), (k)?

7. Also in Fig 1, the authors have not commented about the distinct differences in the ovarian tissue sections of control (a), (d) and (e).

Minor corrections

1. ‘I.U.’ should be consistently presented in upper case throughout the manuscript.

2. Gene and protein names should be consistently represented (italized upper case for genes, lower case for proteins).

3. Spell check manuscript thoroughly, eg. Satined

Reviewer #4: This manuscript investigated the effect of PMSG/hCG induced ovarian stimulation (OS) on the expression of selected Piwi protein genes. Two set of experiments were conducted using either increasing dose of PMSG/hCG or multiple rounds of OS. Number of samples for each treatment group in both experiments were sufficient to allow confidence in the statistical analysis.

The authors concluded that exogenous administration of gonadotrophin led to down-regulation of several Piwi protein genes with suggestion that it was associated with plasma E2 level. This conclusion was most probably based on the fact that the three most down-regulated Piwi protein genes were from the groups with highest level of plasma E2 (Table 3). However, other and more Piwi protein genes were significantly down-regulated in Group 4 in Experiment 1 (Table 3) which had no significant difference in plasma E2 level compared to the control (Table 2). Hence, from the result it seems that only expression level for three of the Piwi protein studied were affected by plasma E2 level. The conclusion should be more specific to prevent misleading the readers.

To make the figure more reader-friendly I would like to suggest the following:

Combine panels i, j and k of Figure 1 to 6 into a single figure showing the localization of the proteins.

Move panels a to h of Figure 1 to 6 to supplementary since the fluorescence intensity data is already presented in the form of bar chart in Figure 7.

Other minor comments:

According to MIQE guidelines (Bustin et al 2009) abbreviation for quantitative PCR should be qPCR to avoid confusion with reverse transcription PCR (RT-PCR).

There are some grammatical errors in the manuscript.

Reviewer #5: The manuscript titled “Effect of ovarian stimulation on the expression of piRNA pathway proteins” is a preliminary expression analysis of piRNA interacting proteins after PMSG/hCG/OS in muse ovary. Though authors fail to draw correlative conclusions and summary with respect to pi RNA biosynthesis other than relative expression.

Specific comments:

-There are many grammatical errors and typos throughout the manuscript.

-Introduction is much focused on general information on the topics in the manuscript and it needs specific input and more correlation why authors have chosen respective treatments to focus their studies on.

-What’s the reason for time variations chosen for OS stimulations?

-The fluorescence images of expression are hard to locate, may need counter stain and arrows/ symbols.

6. PLOS authors have the option to publish the peer review history of their article (what does this mean?). If published, this will include your full peer review and any attached files.

Reviewer #1: No

Reviewer #2: No

Reviewer #3: No

Reviewer #4: No

Reviewer #5: No

---

## [Author Response · Author response to Decision Letter 0]

2 Mar 2020

Response to reviewer and editor comments

(PONE-D-19-34298)

Dear Editor,

First of all, we thank the editor and the reviewers for their thoughtful and supportive comments on our manuscript. We discussed these comments and made some changes on the manuscript according to your recommendations. Our answers to the comments are as follows.

Editor Comments

Comment 1: Rationale of the work, experimental design, choice of doses need to be explained well. The analyses and significant data obtained are limited to gene and protein expression related to piRNA. Hence author should add additional experiments or discuss more about the relevant publications to co-relate the study results to support the conclusion specifically.

Response 1: We reevaluated the discussion section and discussed our results more about the relevant publications according to your recommendations. Thus, a new discussion section was added with new references (marked with red)

Comment 2:

Please ensure that your manuscript meets PLOS ONE's style requirements, including those for file naming. The PLOS ONE style templates can be found at http://www.plosone.org/attachments/PLOSOne_formatting_sample_main_body.pdf and http://www.plosone.org/attachments/PLOSOne_formatting_sample_title_authors_affiliations.pdf

Response 2: We reviewed the manuscript according to PLOS ONE author instructions to meet PLOS ONE’s style requirement.

Comment 3:

We suggest you thoroughly copyedit your manuscript for language usage, spelling, and grammar. If you do not know anyone who can help you do this, you may wish to consider employing a professional scientific editing service. 

Response 3: As recommended by the Editor and the reviewers, we submitted our revised manuscript to the American Journal Experts for professional editing (edited for language usage, spelling, and grammar). A certificate provided from American Journal Experts is attached at the end of this file. A copy of our manuscript showing our changes was uploaded as a "supporting information" file, and a clean copy of the edited manuscript was uploaded as the new “manuscript" file.

Comment 4:

We note that you have included the phrase “data not shown” in your manuscript. Unfortunately, this does not meet our data sharing requirements. PLOS does not permit references to inaccessible data. We require that authors provide all relevant data within the paper, Supporting Information files, or in an acceptable, public repository. Please add a citation to support this phrase or upload the data that corresponds with these findings to a stable repository (such as Figshare or Dryad) and provide and URLs, DOIs, or accession numbers that may be used to access these data. Or, if the data are not a core part of the research being presented in your study, we ask that you remove the phrase that refers to these data.

Response 4: As suggested, phrases that our findings did not concretely support were removed from the manuscript. The removed sentences were in the second paragraph of the discussion section and were marked with grey and “_ “.

Review Comments to the Author:

Reviewer #1: 

Comment 1: As reproductive cycle and age are correlated, why mice of 8-10 weeks were chosen for the study is not explained anywhere in the manuscript.

Comment 2: What’s the reason for dose and time period for hCG and PMSG? Dose and intervals might have induced detrimental effects. There is no strong basis or explanation provided from the authors side in this context. 

Response 1 and 2:

Superovulation of mice is routinely used to increase the number of obtainable oocytes. Overall, 6-12-weeks old age mice can be used for superovulation procedure. In this study, we wanted to investigate the expression of piRNA pathway protein on mature individuals. Doses of 5.0 to 10.0 international units (IU) of PMSG per mouse are very commonly used for superovulation protocols to in the literature [1-3]. That is why, we preferred low-dose 5.0, medium-dose 7.5, high-dose 10.0, and over-dose 12.5 IU PMSG/hCG groups. Furthermore, superovulation by exogenous gonadotropin is a widely used method to produce in vivo‐derived embryos for embryo transfer in women, and in recent years, it has become an important part of IVF. In IVF treatment or reproductive studies, different OS protocols can be used, for instance, sometimes IVF patients or experimental animals used in reproductive studies undergo OS using higher doses of exogenous gonadotrophins to obtain multiple oocytes in a single cycle. Moreover, depending on the success of IVF, some women may undergo repeated ovarian stimulation with exogenous gonadotropins. Therefore, we want to examined whether there is an effect of different doses of exogenous gonadotropin and repeated OS on the expression of piRNA pathway genes and formed study groups for this purpose.

1. Vaseghi H, Mogheiseh A, Sepehrimanesh M, Kafi M, Nooranizadeh MH. Super pregnancy in a BALB/c mouse superovulated with PMSG. Lab Anim Res. 2017;33(3):280–282. doi:10.5625/lar.2017.33.3.280

2. Uysal F, Ozturk S, Akkoyunlu G. Superovulation alters DNA methyltransferase protein expression in mouse oocytes and early embryos. J Assist Reprod Genet. 2018;35(3):503–513. doi:10.1007/s10815-017-1087-z

3. Ozturk S, Yaba-Ucar A, Sozen B, Mutlu D, Demir N. Superovulation alters embryonic poly(A)-binding protein (Epab) and poly(A)-binding protein, cytoplasmic 1 (Pabpc1) gene expression in mouse oocytes and early embryos. ReprodFertil Dev. 2016;28(3):375–383. doi:10.1071/RD14106

4. Combelles, Catherine MH, and David F. Albertini. "Assessment of oocyte quality following repeated gonadotropin stimulation in the mouse." Biology of Reproduction 68.3 (2003): 812-821

Comment 3 Why FSH and LH levels, body weights were not measured as a control for ovarian stimulation occurance.

Comment 10. It has been said in the conclusion that genome instability occurs on external gonadotrophin administration depending on the dose and the repetition of administration. Did author test on any genome instability that arises out of disrupted piRNA biogenesis (for eg microsatellite) detection like using PCR were done on the tissues?

Response 3 and 10

We appreciate the reviewer’s comments on the above concern. However, the financial support of this project was limited to conducting relevant experiments only for the purpose of this research. In order to make more detailed investigations, we are currently working on the preparation of a grant proposal that focuses on the piRNA pathway and high-throughput sequencing of piRNAs after the treatment of different doses of exogenous gonadotropin and repeated rounds of OS. While planning this project, your valuable suggestions will be taken into consideration.

Comment 4: The exact function of Mili, Miwi, MaeI, Tdrd1, Tdrd9, Mitopld genes in the piRNA pathway is not very well explained in the manuscript.

Comment 12. The molecular mechanisms underlying piRNA biogenesis and functions are complex and diverse, the paper must provide more emphasis on step by step piRNA biogenesis and overall biological impact of the study.

Response 4 and 12: We agree with the comment and have added 4 paragraphs in the introduction (paragraphs 4, 5, 6 and 7 in the 'Revised Manuscript with Track Changes' file) to better explain the functions of these genes and the piRNA biogenesis. Taking into account the suggestions, the introduction was revised; new sentences have been added and some sentences have been deleted (marked with purple and “_ “).

Comment 5: The authors should consider making an illustration or model figure for piRNA pathway and corresponding changes upon ovarian stimulation.

Response 5: We appreciate the reviewer’s constructive comments. We believe that this will also contribute to easier understanding of the piRNA biogenesis and the purpose and result of the study. As suggested by the reviewer, we added an overview figure (S7 Fig) for piRNA pathway to the supporting information.

Comment 6: The authors should present qPCR results in the form of a bar /column chart.

Response 6: As suggested by the reviewer, we presented qPCR results in the form of a bar/column chart (Fig 1).

Comment 7: The manuscript carries typos and grammatical errors at various places which has to be keenly addressed For eg. in abstract line 9 “ remarkable decreases”, Page 2 line 10 “piRANA”, Page 9 , Page 19 “was found that” and reference 6 “ editor” etc. Hence it has to be proof read.

Response 7: We apologize for the mistakes in line 9, page 2 line 10 and page 19 and reference 6. Typos and grammatical errors in these sections were corrected. Furthermore, as recommended by the Editor and the reviewers, we submitted our revised manuscript to the American Journal Experts for professional editing (edited for language usage, spelling, and grammar). 

Comment 8. The images are not very clear; hence a zoomed portion in the tissue section has to be included.

Response 8: As per reviewer’s advice, high quality figures with 30-600 dpi resolution were produced and incorporated in the manuscript.

Comment 9: Image analysis can be well explained separately in the methods.

Response 9: Thank you for this advice. As the reviewer stated, we revised Immunofluorescence labeling in the methods and added a subtitle called “Image analysis” with a brief detailing of this analysis.

Comment 11: It has been found that piRNA has different roles in cytoplasm and nucleus. The author has to experimentally or theoretically address pathway in which compartment Mili, Miwi, MaeI, Tdrd1, Tdrd9, Mitopld genes are specifically affected?

Response 11: In paragraphs 3, 4 and 5, sentences containing information about the cellular compartments or locations of these proteins were added (e.g. miwi and mili are cytosolic proteins- miwi2 is a nuclear protein- mitochondrial protein mitopld- mael is a nucleo-cytoplasmic shuttling protein- tudor proteins are localized in cytoplasmic granules termed “nuages”).

Reviewer #2: 

Comment 1: How were dose ranges for PMSG and hCG decided? Would the authors see difference in protein expression levels in other dose concentrations?

Response 1: Superovulation of mice is routinely used to increase the number of obtainable oocytes. Overall, 6-12-weeks old mice can be used for superovulation. In this study, we wanted to investigate the expression of piRNA pathway protein on mature individuals. Doses of 5.0 to 10.0 international units (IU) of PMSG per mouse are very commonly used for superovulation protocols in the literature [1-3]. That is why we preferred low-dose 5.0, medium-dose 7.5, high-dose 10.0, and over-dose 12.5 IU PMSG groups.

1. Vaseghi H, Mogheiseh A, Sepehrimanesh M, Kafi M, Nooranizadeh MH. Super pregnancy in a BALB/c mouse superovulated with PMSG. Lab Anim Res. 2017;33(3):280–282. doi:10.5625/lar.2017.33.3.280

2. Uysal F, Ozturk S, Akkoyunlu G. Superovulation alters DNA methyltransferase protein expression in mouse oocytes and early embryos. J Assist Reprod Genet. 2018;35(3):503–513. doi:10.1007/s10815-017-1087-z

3. Ozturk S, Yaba-Ucar A, Sozen B, Mutlu D, Demir N. Superovulation alters embryonic poly(A)-binding protein (Epab) and poly(A)-binding protein, cytoplasmic 1 (Pabpc1) gene expression in mouse oocytes and early embryos. ReprodFertil Dev. 2016;28(3):375–383. doi:10.1071/RD14106

Comment 2: Pictorial summarization/representation of the observed data pathway would help comprehend the data with ease.

Response 2: Thank you for this remark. As suggested by the reviewer, we added an overview figure for of the observed data pathway to the supporting information. We believe that this will also contribute to easier understanding of the piRNA biogenesis and the purpose and result of the study. 

Comment 3. Immunofluorescence data could be better represented by enlarged images.

As per the reviewer’s advice, immunofluorescence images were revised, and high quality figures with 30-600 dpi resolution were produced and incorporated in the manuscript.

Comment 4. Typos and grammatical errors are abundant. For eg. in abstract line 5, “ 7.5 PMSG/hCG” – IU is missing, page 2 line 4, “ embryo and offspring elicited by OS”, page 2 line 15 “transposons, regulation of some gene expression”, page 6 line 21, “investigated in our study examined in MII oocytes”, page 7 line 18 and 20 “ 1.u”, page 7 line 29 “caused to an increase”, page 9 line 19 “ Kabayama et al…..retrotransposon transcripts [36]”- please rephrase this sentence.

Response 4: We sincerely apologize for these mistakes. Typos and grammatical errors in these sections were corrected. Furthermore, as recommended by the Editor and the reviewers, we submitted our revised manuscript to the American Journal Experts for professional editing (edited for language usage, spelling, and grammar).

Comment 5: Please pay attention to Author contribution in page 13. Who is J.L?

Response 5: We apologize for the mistake in the author contribution section, J.L was deleted from this section.

Reviewer #3: 

Introduction and methods:

Comment 1. The phrase ‘piRNA pathway’ is not clearly defined, whether it refers to biogenesis of piRNA or functions hereafter. Also, piRNA and hormones should be elaborated with appropriate references.

Response 1: Thank you for this advice. The following sentence, which clearly defines the piRNA pathway, was added to the introduction.

“The piRNA pathway, which contains PIWI proteins, piRNAs and proteins that have a role in piRNA biogenesis, can maintain genome integrity by transposon silencing. Furthermore, the references of the manuscript were revised. Some of the references were removed, and a few new references were added.

Comment 2. What is the difference between ‘controlled OS’ and ‘OS’? it is not clear what was followed in the experimental setup as the terms are used interchangeably in the manuscript. ?

Response 2: We deeply apologize for the confusion. In fact, controlled OS and OS were used with the same meaning. To avoid confusion, only the term“OS” was used throughout the article.

Comment 3. Immunofluorescence labelling in methods is not clear. “After the sections were heated in 10 mM……. phosphate-buffered saline with tween-20 (PBS-Tween-20)” – consider simplifying and rephrasing this sentence. 

Response 3:

We agree with the comment and have revised Immunofluorescence labelling section and appropriate sentences. We hope the current form is clearer.

Results:

Comment 1: It is not clear why the authors compare the results of both the experiments 1 and 2 together only in the immunoexpression studies. 

Response 1: In qPCR and immunofluorescence analysis, the data obtained from the study groups were compared with the controls. For a brief and understandable presentation, the results of experiment 1 and experiment 2 were given on the same bar graph for each protein or gene.

Comment 2: Table 3 results would be better if represented as a bar graph to highlight the differential expression levels.

Response 2: Thank you for this advice. As suggested by the reviewer, we presented qPCR results in the form of a bar/column chart (Fig 1)

Comment 3: The sentence “

Based on this studies and the obtained data from the ……………………………………and these increasing rates were similar in groups 1, 2 and 3” needs clarification. The authors have not explained the relationship between the different hormone injections and their respective effect on receptors. Rather they have just made statements about hormone injections and receptors with cited references. As mentioned by the authors, the relationship between E2, FSH, LH and the receptor regulation of these hormones is complex and requires valid experimental evidence. The authors should include experimental data on the expression levels of these receptors under the current experimental conditions to make a relevant connection.

Comment 4: What is the quality of the oocyte and embryo developed under the tested experimental conditions. This result along with the data on FSH, LH, E2 hormone/receptor levels will help to connect the dots. 

Response 3 and 4: We appreciate the reviewer’s comments on the above concern. Phrases that our findings did not concretely support were removed from the manuscript. The removed sentences were in the second paragraph of the discussion section and were marked with grey and “_ “. We fully agree with the reviewer's opinion about FSH, LH, E2 hormone/receptor. However, the financial support of this project was limited to conducting relevant experiments only for the purpose of this research. In order to make more detailed investigations, we are currently working on the preparation of a grant proposal that focuses on the piRNA pathway and high-throughput sequencing of piRNAs after the treatment of different doses of exogenous gonadotropin and repeated rounds of OS. While planning this project, your valuable suggestions will be taken into consideration.

Comment 5: Why are expression levels of Tdrd 9, Tdrd1, and Mael decreased in 2R but not in 3R and 4R. There seems to be a positive correlation between the rounds of OS and the rescue in expression levels compared to the control. What is the reason for this? 

Response 5: Indeed, there was a significant decrease in the expression levels of these 3 genes in the 3R group. A decrease in the 4 R group was observed, but it did not reach a significant value. The highest reduction rates were observed in the 2R group. Observing parallel decrease rates in E2 levels in the same groups may indicate that the expression of these 3 genes is suppressed with high E2 levels, as we mentioned in the manuscript.

Comment 6: Fig1 legend is not clear. What is control (a) and control (j), (k)?

Comment 7: Also in Fig 1, the authors have not commented about the distinct differences in the ovarian tissue sections of control (a), (d) and (e). 

Response 6 and 7: We apologize for the mistakes in Fig1 legends. We revised the figures and figure legends. In addition, high quality figures with 300-600 dpi resolution were produced and incorporated in the manuscript. Furthermore, as a result of image analysis, the immunofluorescence data that reached statistical significance was mentioned in the results and discussion sections.

Minor corrections

Comment 1 ‘I.U.’ should be consistently presented in upper case throughout the manuscript. 

Response 1: I.U.was presented in upper case throughout the manuscript

Comment 2. Gene and protein names should be consistently represented (italized upper case for genes, lower case for proteins).

Response 2: Thanks for your attention and suggestion. As suggested, gene symbols were italicized, all letters were typed in upper case, and protein names were typed in lower case but not italicized.

Comment 3: Spell check manuscript thoroughly, eg. Satined 

Response 3: We apologize for this situation.We spell-checked the manuscript and cleaned up the stained areas.

Reviewer #4: 

Comment 1: The authors concluded that exogenous administration of gonadotrophin led to down-regulation of several Piwi protein genes with suggestion that it was associated with plasma E2 level. This conclusion was most probably based on the fact that the three most down-regulated Piwi protein genes were from the groups with highest level of plasma E2 (Table 3). However, other and more Piwi protein genes were significantly down-regulated in Group 4 in Experiment 1 (Table 3) which had no significant difference in plasma E2 level compared to the control (Table 2). Hence, from the result it seems that only expression level for three of the Piwi protein studied were affected by plasma E2 level. The conclusion should be more specific to prevent misleading the readers. 

Response 1: We intensely appreciate the reviewer’s attention and comment. We fully agree, and we apologize for our sentences that caused misunderstanding on this issue. We revised these sentences in the abstract and conclusion sections as to mean that there could be a relationship only between E2 levels and Tdrd9, Tdrd1 and Mael expression levels.

Comment 2.

To make the figure more reader-friendly I would like to suggest the following:

Combine panels i, j and k of Figure 1 to 6 into a single figure showing the localization of the proteins. Move panels a to h of Figure 1 to 6 to supplementary since the fluorescence intensity data is already presented in the form of bar chart in Figure 7

Response 2: Thank you for these constructive comments. As suggested, we revised the figures, and high quality figures with 300-600 dpi resolution were produced and incorporated in the manuscript.

Comment 3: According to MIQE guidelines (Bustin et al 2009) abbreviation for quantitative PCR should be qPCR to avoid confusion with reverse transcription PCR (RT-PCR).

Response 3: Thanks for your advice. We changed qRT-PCR to qPCR in the whole manuscript.

Comment 4. There are some grammatical errors in the manuscript.

Response 4: Typos and grammatical errors were corrected. Furthermore, as recommended by the Editor and the reviewers, we submitted our revised manuscript to the American Journal Experts for professional editing (edited for language usage, spelling, and grammar).

Reviewer #5: 

Specific comments:

Comment 1-There are many grammatical errors and typos throughout the manuscript.

Response 1: Typos and grammatical errors were corrected. Furthermore, as recommended by the Editor and the reviewers, we submitted our revised manuscript to the American Journal Experts for professional editing (edited for language usage, spelling, and grammar).

Comment 2-Introduction is much focused on general information on the topics in the manuscript and it needs specific input and more correlation why authors have chosen respective treatments to focus their studies on.

Response 2: We appreciate the reviewer's comments on our manuscript. We agree with the comment, and as suggested, we added new sentences and paragraphs in the introduction (paragraphs 4, 5, 6 and 7 in the 'Revised Manuscript with Track Changes' file) to better explain the functions of investigated genes and the aim of this study. Furthermore, superovulation by exogenous gonadotropin is a widely used method to produce in vivo‐derived embryos for embryo transfer in women, and in recent years, it has become an important part of IVF. In IVF treatment or reproductive studies, different OS protocols can be used, for instance, sometimes IVF patients or experimental animals used in reproductive studies undergo OS using higher doses of exogenous gonadotrophins to obtain multiple oocytes in a single cycle. Moreover, depending on the success of IVF, some women may undergo repeated ovarian stimulation with exogenous gonadotropins. Therefore, we wanted to examine whether there is an effect of different doses of exogenous gonadotropin and repeated OS on the expression of piRNA pathway genes and formed study groups for this purpose.

Comment 3-What’s the reason for time variations chosen for OS stimulations? 

Response 3: In some cases, IVF procedure can be repeated due to failure of implantation or pregnancy. That is why we wanted to mimic the clinic conditions and show the interactions between superovulation and piRNA pathway proteins.

Comment 4-The fluorescence images of expression are hard to locate, may need counter stain and arrows/ symbols.

Response 4: Thank you for these constructive comments. As suggested, we revised the figures, and high quality figures with 300-600 dpi resolution were produced and incorporated in the manuscript.

---

## [Decision Letter · Decision Letter 1]

24 Mar 2020

PONE-D-19-34298R1

Effect of ovarian stimulation on the expression of piRNA pathway proteins

PLOS ONE

Dear Dr. Sari,

Thank you for submitting your manuscript to PLOS ONE. After careful consideration, we feel that it has merit but does not fully meet PLOS ONE’s publication criteria as it currently stands. Therefore, we invite you to submit a revised version of the manuscript that addresses the points raised during the review process.

The manuscript is way improved, however still few errors are seen and few more points need to be explained in detail. Authors should edit the manuscript for Grammar and Plos style and also address all the comments of reviewers carefully.

We would appreciate receiving your revised manuscript by May 08 2020 11:59PM. To enhance the reproducibility of your results, we recommend that if applicable you deposit your laboratory protocols in protocols.io, where a protocol can be assigned its own identifier (DOI) such that it can be cited independently in the future. For instructions see: http://journals.plos.org/plosone/s/submission-guidelines#loc-laboratory-protocols

We look forward to receiving your revised manuscript.

Kind regards,

Academic Editor

PLOS ONE

Additional Editor Comments (if provided):

The manuscript is way improved, however still few errors are seen and few more points need to be explained in detail. Authors should edit the manuscript for Grammar and Plos style and also address all the comments of reviewers carefully.

Reviewers' comments:

Reviewer's Responses to Questions

**Comments to the Author**

1. If the authors have adequately addressed your comments raised in a previous round of review and you feel that this manuscript is now acceptable for publication, you may indicate that here to bypass the “Comments to the Author” section, enter your conflict of interest statement in the “Confidential to Editor” section, and submit your "Accept" recommendation.

Reviewer #1: (No Response)

Reviewer #3: (No Response)

Reviewer #4: All comments have been addressed

2. Is the manuscript technically sound, and do the data support the conclusions?

Reviewer #1: Partly

Reviewer #3: Partly

Reviewer #4: Yes

3. Has the statistical analysis been performed appropriately and rigorously? 

Reviewer #1: Yes

Reviewer #3: Yes

Reviewer #4: Yes

4. Have the authors made all data underlying the findings in their manuscript fully available?

Reviewer #1: Yes

Reviewer #3: Yes

Reviewer #4: Yes

5. Is the manuscript presented in an intelligible fashion and written in standard English?

Reviewer #1: Yes

Reviewer #3: Yes

Reviewer #4: Yes

6. Review Comments to the Author

Reviewer #1: The manuscript has been changed partially as per the reviewer’s comments however, the manuscript needs come structural change and major corrections to make it more easy for the readers.

1. Figure -1 - Why the Error bars are missing ?

2. mili, miwi, mael, tdrd1, tdrd9 gene name/ protein names should be written in corresponding Capital or italics

3. Figure legends has to be detailed with subdivisions a,b, ... described and legends has to be seperate and Results and Discussion should not merely describe the numerical data. More of scientific impact of the study has to be written

4. Less coherent information is important than more incoherent information, Hence kindly make the manuscript as simple as possible.

5. S7 has to be incorporated as main figure with complete explanation of the pathway as figure legends

6. Authors need to see other plos one papers more carefully for better formatting of the manuscript.

7. Tables has to be separate not in the text.

8. Still needs grammar improvement - For eg In abstract line 23 - "greatest decreases " decrease

Reviewer #3: The authors have addressed all my comments, though not adequately.

The response to comment 5 was not satisfactory. The authors have mentioned that expression levels of Tdrd 9, Tdrd1, and Mael were less in 3R and 4R (not as significant) compared to control but 2R showed maximum suppression in expression. This data correlates with E2 levels. But the authors have not given an explanation for the observation of why gene expression is low in 2R than in 3R OR 4R. I am curious to understand if increase in repetitive rounds of OS is better than 2R. It is interesting to find decrease in E2 levels in 3R and 4R compared to 2R, what is the most logical explanation for this?

L 258: new abbreviation 'FR' should be mentioned first in L 256

Reviewer #4: (No Response)

7. PLOS authors have the option to publish the peer review history of their article (what does this mean?). If published, this will include your full peer review and any attached files.

Reviewer #1: No

Reviewer #3: No

Reviewer #4: No

---

## [Author Response · Author response to Decision Letter 1]

4 Apr 2020

Dear Editor,

We thank you and the reviewers for the thorough reading and constructive comments of our manuscript. We are pleased to submit the improved research article, for your consideration in the PLOS ONE. Our response to the editor and reviewer comments is as follows.

Editor Comments

Comment: The manuscript is way improved, however still few errors are seen and few more points need to be explained in detail. Authors should edit the manuscript for Grammar and Plos style and also address all the comments of reviewers carefully.

Response: Thank you for the comment. We carefully reviewed the manuscript again according to PLOS ONE author instructions to meet PLOS ONE’s style requirement. We have resubmitted our revised manuscript to a professional editing firm (Protranslate) for professional grammar editing. The changes are marked in red (blue in pdf file). A certificate provided from Protranslate is attached at the end of this file. A copy of our manuscript showing our changes was uploaded as a "'Revised Manuscript with Track Changes'" file, and a clean copy of the edited manuscript was uploaded as the new “manuscript" file.

Review Comments to the Author:

Reviewer #1: 

Comment 1:

Figure -1 - Why the Error bars are missing?

Response 1: We apologize for this mistake. We have added the error bars in the Fig 2 (updated caption).

Comment 2: mili, miwi, mael, tdrd1, tdrd9 gene name/ protein names should be written in corresponding Capital or italics

Response 2: Thank you for this advice. We updated the gene and protein symbols according to “MGI-Guidelines for Nomenclature of Genes, Genetic Markers, Alleles, & Mutations in Mouse & Rat” (Gene symbols generally are italicized, with only the first letter in uppercase and the remaining letters in lowercase. Protein designations are the same as the gene symbol, but are not italicized and all are upper case). 

Comment 3: Figure legends has to be detailed with subdivisions a,b, ... described and legends has to be seperate and Results and Discussion should not merely describe the numerical data. More of scientntroduction, results and discussion section. 

Response 3: We agree with the comment. As suggested, we revised the figure legends and citation of figures in the introduction, results, and discussion sections (Marked with pink).

Comment 4: Less coherent information is important than more incoherent information, Hence kindly make the manuscript as simple as possible.

Response 4: Thank you for the suggestion. We apologize for our confusing statement. Because some of the statements were confusing and/or redundant, we have removed (marked with green) or revised some of them (marked with yellow). 

Comment 5: S7 has to be incorporated as main figure with complete explanation of the pathway as figure legends.

Response 5: We appreciate your suggestion. As per the reviewer’s advice, we have incorporated Fig S7 as the main figure (Fig 1) with a complete explanation of the pathway as figure legends and revised partially. We hope the current form will be clearer.

Comment 6: Authors need to see other plos one papers more carefully for better formatting of the manuscript.

Response 6: Thank you for the comment. We have adapted the manuscript according to PLOS ONE's requirements.

Comment 7: Tables has to be separate not in the text.

Response 7: Thank you for the comment, but according to the submission guidelines, each table has to be placed in the manuscript file directly after the paragraph in which it is first cited. Tables must not be submitted in the separate files.

Comment 8: Still needs grammar improvement - For eg In abstract line 23 - "greatest decreases " decrease

Response 8: We deeply apologize for grammatical errors. Grammatical error in line 23 has been corrected. Furthermore, we have resubmitted our revised manuscript to a professional editing firm (Protranslate) for professional grammar editing. The changes are marked in red (blue in pdf file).. A certificate provided from Protranslate is attached at the end of this file.

Reviewer #3: 

Comment 1: The response to comment 5 was not satisfactory. 

Response 1: We are sorry for that comment. In order to create a more suitable figure, we have incorporated Fig S7 as the main figure (Fig 1) with a complete explanation of the pathway as figure legends and revised partially. We hope the current form will be clearer.

Comment 2: The authors have mentioned that expression levels of Tdrd 9, Tdrd1, and Mael were less in 3R and 4R (not as significant) compared to control but 2R showed maximum suppression in expression. This data correlates with E2 levels. But the authors have not given an explanation for the observation of why gene expression is low in 2R than in 3R OR 4R. I am curious to understand if increase in repetitive rounds of OS is better than 2R. It is interesting to find decrease in E2 levels in 3R and 4R compared to 2R, what is the most logical explanation for this?

Response 2: We appreciate the reviewer's comments on the subject above. We agree broadly with your consideration. Indeed, there was a significant decrease in the expression levels of these 3 genes in the 3R group. But as mentioned, the reduction rate in 2R was the highest. This highest reduction rate observed in the expression of Tdrd 9, Tdrd1 and Mael in the 2R group correlated with E2 levels suggests that there may be a complex regulation mechanism between the genes we examined and the other genes involved in piRNA biogenesis or other factors that have the potential to affect the E2 level in the repeated OS process. At first glance, increasing the number of OS cycles appears to have a positive effect on at least these 3 genes. However, to reveal the main effects of OS on this pathway in detail, we think that in different procedures further studies are needed to examine piRNA sequencing, FSH, LH, E2 hormone and receptor levels, and expression and the function of the proteins involved in the piRNA pathway. Furthermore, as we mentioned in the discussion section of our manuscript, all these findings suggest that the effect of plasma E2 levels on the expression of these proteins may vary from protein to protein in the piRNA pathway or there may be a complex regulatory pattern between these proteins.

Comment 3: L 258: new abbreviation 'FR' should be mentioned first in L 256

Response 3: We apologize for the mistake in line no. 256. We have mentioned abbreviation 'FR' (Fold regulation).

---

## [Decision Letter · Decision Letter 2]

15 Apr 2020

PONE-D-19-34298R2

Effect of ovarian stimulation on the expression of piRNA pathway proteins

PLOS ONE

Dear Dr. Sari,

Thank you for submitting your manuscript to PLOS ONE. After careful consideration, we feel that it has merit but does not fully meet PLOS ONE’s publication criteria as it currently stands. Therefore, we invite you to submit a revised version of the manuscript that addresses the points raised during the review process.We would appreciate receiving your revised manuscript by May 30 2020 11:59PM. To enhance the reproducibility of your results, we recommend that if applicable you deposit your laboratory protocols in protocols.io, where a protocol can be assigned its own identifier (DOI) such that it can be cited independently in the future. For instructions see: http://journals.plos.org/plosone/s/submission-guidelines#loc-laboratory-protocols

We look forward to receiving your revised manuscript.

Kind regards,

Rajakumar Anbazhagan, Ph. D.

Academic Editor

PLOS ONE

Additional Editor Comments (if provided):

Manuscript is acceptable pending minor corrections.

Reviewers' comments:

Reviewer's Responses to Questions

**Comments to the Author**

1. If the authors have adequately addressed your comments raised in a previous round of review and you feel that this manuscript is now acceptable for publication, you may indicate that here to bypass the “Comments to the Author” section, enter your conflict of interest statement in the “Confidential to Editor” section, and submit your "Accept" recommendation.

Reviewer #1: All comments have been addressed

Reviewer #3: All comments have been addressed

2. Is the manuscript technically sound, and do the data support the conclusions?

Reviewer #1: Partly

Reviewer #3: Yes

3. Has the statistical analysis been performed appropriately and rigorously? 

Reviewer #1: Yes

Reviewer #3: Yes

4. Have the authors made all data underlying the findings in their manuscript fully available?

Reviewer #1: Yes

Reviewer #3: Yes

5. Is the manuscript presented in an intelligible fashion and written in standard English?

Reviewer #1: Yes

Reviewer #3: (No Response)

6. Review Comments to the Author

Reviewer #1: 1-Line number 117- decreasing symbols has to be removed. Punctuation has to be checked.

2-MITOPLD, MILI, TDRD..etc protein names has to be consistent through out the manuscript when it comes to deciding uppercase or lower case.

3-Figure -1 needs figure legends

4-In Figure-1 typo “ Secondery”

Reviewer #3: Few suggestions: check the use of articles and punctuation in figure 1 legend. Also cite references for explaining fig1.

7. PLOS authors have the option to publish the peer review history of their article (what does this mean?). If published, this will include your full peer review and any attached files.

Reviewer #1: No

Reviewer #3: No

---

## [Author Response · Author response to Decision Letter 2]

16 Apr 2020

Response to reviewer and editor comments

(PONE-D-19-34298R2)

Dear Editor,

First of all, we would like to thank the editor and all of the reviewers for their careful review of the manuscript and professional comments and suggestions for improving our initial manuscript. We are pleased to submit the improved research article, for your consideration in the PLOS ONE. Our response to the reviewer's comments is as follows.

Reviewer #1: 

Comment 1-Line number 117- decreasing symbols has to be removed. Punctuation has to be checked.

Response 1-Thank you for this advice. As your advice, line number 117- decreasing symbols has been removed from the manuscript. Punctuation has been checked and some corrections made in the manuscript (marked with red).

Comment 2-MITOPLD, MILI, TDRD..etc protein names has to be consistent through out the manuscript when it comes to deciding uppercase or lower case.

Response 2- We deeply apologize for our inattention that caused inconsistency about this issue. We carefully reviewed the manuscript again according to MGI-guidelines for the nomenclature of genes and proteins. We made some corrections in the manuscript, and figure 2 according to the MGI-guidelines.

Comment 3-Figure -1 needs figure legends.

Response 3- Thanks for your suggestion. As suggested, Fig.1 legends have added to the manuscript.

Comment 4-In Figure-1 typo “ Secondery”

We apologize for this misspelling. We have corrected this word in the Fig.1

Reviewer #3: : 

Comment 1-Few suggestions: check the use of articles and punctuation in figure 1 legend. Also cite references for explaining fig1.

Response 1- Thanks for your suggestion. The punctuation in Fig.1 legends and the manuscript were rechecked and corrections were made (marked with red). Furthermore, references have been added for the description of fig 1.

---

## [Editor Report · Decision Letter 3]

20 Apr 2020

Effect of ovarian stimulation on the expression of piRNA pathway proteins

PONE-D-19-34298R3

Dear Dr. Sari,

We are pleased to inform you that your manuscript has been judged scientifically suitable for publication and will be formally accepted for publication once it complies with all outstanding technical requirements.

With kind regards,

Rajakumar Anbazhagan, Ph. D.

Academic Editor

PLOS ONE
---

## [Editor Report · Acceptance letter]

23 Apr 2020

PONE-D-19-34298R3 

Effect of ovarian stimulation on the expression of piRNA pathway proteins 

Dear Dr. Sari:

I am pleased to inform you that your manuscript has been deemed suitable for publication in PLOS ONE. Congratulations! Your manuscript is now with our production department. 

With kind regards,

on behalf of

Dr. Rajakumar Anbazhagan 

Academic Editor

PLOS ONE